# Imbalanced machine learning classification models for removal biosimilar drugs and increased activity in patients with rheumatic diseases

**David Castro Corredor**[1]*, **Luis Ángel Calvo Pascual**[2]

**1** Rheumatology Department, Hospital General Universitario de Ciudad Real, Ciudad Real, Spain,
**2** Department of Quantitative Methods. ICADE, Universidad Pontificia de Comillas Madrid, Madrid, Spain

* d.castrocorredor@gmail.com

## Abstract

### Objective

Predict long-term disease worsening and the removal of biosimilar medication in patients with rheumatic diseases.

### Methodology

Observational, retrospective descriptive study. Review of a database of patients with immune-mediated inflammatory rheumatic diseases who switched from a biological drug (biosimilar or non-biosimilar) to a biosimilar drug for at least 6 months. We selected the most important variables, from 18 variables, using mutual information tests. As patients with disease worsening are a minority, it is very difficult to make models with conventional machine learning techniques, where the best models would always be trivial. For this reason, we computed different types of imbalanced machine learning models, choosing those with better f1-score and mean ROC AUC.

### Results

We computed the best-imbalanced machine learning models to predict disease worsening and the removal of the biosimilar, with f1-scores of 0.52 and 0.63, respectively. Both models are decision trees. In the first one, two important factors are switching of biosimilar and age, and in the second, the relevant variables are optimization and the value of the initial PCR.

### Conclusions

Biosimilar drugs do not always work well for rheumatic diseases. We obtain two imbalanced machine learning models to detect those cases, where the drug should be removed or where the activity of the disease increases from low to high. In our decision trees appear not previously studied variables, such as age, switching, or optimization.

**Data Availability Statement:** All relevant data are within the paper and its Supporting Information files.

**Funding:** The authors received no specific funding for this work.

**Competing interests:** The authors have declared that no competing interests exist.

## Introduction

Biological drugs (bDMARDs) have revolutionized the conventional treatment of inflammatory rheumatic diseases, significantly improving the quality of life for our patients, both in terms of joint and extra-articular outcomes [1]. Their main drawback, the economic cost, can be alleviated using biosimilars [2]. A biosimilar is a biological medicine that contains a version of the active substance found in a previously authorized original biological medicine (reference medicine). Similarity to reference medicine must be established through a comparability exercise regarding quality characteristics, biological activity, safety, and efficacy [3]. During their approval process, biosimilars have demonstrated to European and American drug agencies that the present variability and any differences from the original drug do not affect safety and efficacy [2, 3]. These studies are designed to optimize the opportunity to detect clinical differences between biosimilars and reference products in homogeneous populations but do not reflect the use of biosimilars in daily practice with a heterogeneous population with associated comorbidities [4]. Given the limited clinical experience with biosimilar use, the importance of pharmacovigilance is emphasized in the drug information leaflets [4, 5].

In the field of rheumatology, the quest for more effective diagnostic and prognostic tools has always been a paramount concern, given the complexity and heterogeneity of rheumatic diseases. Over the years, traditional statistical models have provided valuable insights into patient outcomes and disease progression. However, with the advent of machine learning (ML), a new era of innovation has emerged, revolutionizing the landscape of rheumatological research and clinical practice. [6–8].

The widespread adoption of conventional ML models, such as support vector machines, decision trees, and random forests, in rheumatology has encountered limitations when dealing with imbalanced data. Such models prioritize overall accuracy, which may yield seemingly satisfactory results when the majority class is correctly classified. However, these models tend to neglect the minority class, leading to poor performance in identifying crucial cases of rare rheumatological conditions. Misclassifying such instances can have severe consequences for patients, delaying accurate diagnoses and appropriate treatments.

According to the literature, the number of patients who worsen after switching to biosimilar drugs is very low, around 7% [9]. Ordinary statistical techniques and conventional machine learning cannot detect these few patients because, in these cases, the best model would always be the trivial one, i.e. the model considering that all patients do not worsen, with a validity of 93%. Imbalanced machine learning models aim to mitigate the biases caused by class imbalances and improve the overall performance in identifying the minority class or classes of interest. Unlike conventional ML, where one can perform Bayesian optimization to obtain the model with the best accuracy in a finite space of models and hyperparameters, in the case of imbalanced ML models, there is no method to choose the best model for the dataset. The methodology, in this case, consists of comparing different types of oversampling, undersampling, and weighted ensemble models, using the f1-score to measure the prediction power of patients who worsen and the area under the ROC curve (AUC) and the accuracy of the confusion matrix as metrics for evaluating the accuracy of the model. [10].

In this paper, we analyze nine different types of imbalanced machine learning classification models, varying their weights and hyperparameters to obtain the best model predicting the long-term deterioration of rheumatic diseases in patients and the removal of the biosimilar. We obtained two decision trees, where emerges non-expected variables such as age as a relevant factor explaining why a few patients have small activity at 16 weeks and high disease activity at 24 weeks. We also pointed out the importance of switching and optimization to explain the removal of the biosimilar.

## Materials and methods

### Study design

This study is an observational and descriptive study. We are considering a retrospective review of a database of patients with inflammatory immune-mediated rheumatic diseases who underwent a prior biologic switch to a biosimilar drug. During the months of November 2022 to February 2023, information was collected on patients with immune-mediated rheumatic diseases treated with biosimilar drugs attended in the outpatient clinics of the Rheumatology Service of the General University Hospital of Ciudad Real in the period in which the study was carried out. interchangeability of the reference drug to the biosimilar.

### Patients

The study includes patients with inflammatory immune-mediated rheumatic diseases, such as predominantly axial spondyloarthritis (radiographic and non-radiographic axial spondyloarthritis) and predominantly peripheral spondyloarthritis (psoriatic arthritis, reactive arthritis, spondyloarthritis associated with inflammatory bowel disease, undifferentiated spondyloarthritis) based on ASAS 2009 criteria, rheumatoid arthritis based on EULAR 2010 criteria, and other rheumatic inflammatory diseases like systemic lupus erythematosus, Behçet's disease, Sjögren's syndrome, myopathies, and syndrome from PAPA (Pyogenic Arthritis, Pyoderma Gangrenosum, and Acne). Patients received treatment during outpatient visits at the Rheumatology Department of the General University Hospital of Ciudad Real for at least 24 weeks from December 2021 to June 2022. Participants who met the inclusion criteria and once informed about the study and after having signed the informed consent were included in the study.

### Variables

The collected variables include demographic data (sex and age) and information on the diseases studied. We collected data on the biosimilar biologic drug used (infliximab, etanercept, adalimumab, and rituximab), whether and which concomitant conventional DMARDs were used, and the patients' associated comorbidities. Additionally, disease activity variables were collected two times, at 16 weeks and at 24 weeks, as follows:

- For patients diagnosed with axial spondyloarthritis and psoriatic arthritis with axial involvement we use the ASDAS-CRP (Ankylosing Spondylitis Activity Score), which includes both subjective variables (e.g., questions about spinal pain, global assessment of the patient, peripheral pain or swelling, or duration of stiffness) and an objective variable of inflammation (CRP). Disease activity was classified as inactive when the score was <1.3, moderate if 1.3–2.1, high if 2.1–3.5, and very high if >3.5.

- For patients with psoriatic arthritis, the DAPSA index (Disease Activity for Psoriatic Arthritis) was used, computed by combining five variables: number of swollen joints, number of tender joints, pain measured using a 0–10 visual numeric scale (VNS), patient global assessment using a 0–10 VNS, and CRP (mg/dl).

- For patients with rheumatoid arthritis, the DAS28-CRP index was used, calculated based on the 28-joint score (joint pain and inflammation), CRP, and the patient's subjective assessment of their pain level. Disease activity was categorized as inactive if the score was <2.6, low activity if 2.6–3.2, moderate activity if 3.2–5.1, and high activity if >5.1. Furthermore, the acute phase reactants ESR (mm/1h) and CRP (mg/dl) were measured. Other variables

related to biosimilar DMARDs, such as drug survival, optimization, reasons for discontinuation, and adverse events, were also assessed.

Moreover, in this paper, the imbalanced target variables are:

- "worsening". It is a binary variable defined as follows: it is one if the activity at 16 weeks is 0 and the activity at 24 weeks is 1, otherwise, it is zero.

- "removal". It is a binary variable that is one when the biosimilar is removed not only because the activity not decreases at 24 weeks but also it is carried out due to the side effects in the patient.

- "optimization". It is a binary variable defined as the use of fewer drug doses per improvement in patient activity.

## Data analysis

To select the most important variables, we employed the mutual information test because, unlike other feature selection methods, it captures linear, nonlinear, and complex relationships because it measures the difference between the joint distribution between variables and marginal distribution using the Kullback-Leibler divergence function. To avoid possible noise when executing the test only once, we executed it 1000 times and taken the mean of the mutual information values of the variables (**Fig 1**). We employed the *sklearn* library (*Python*).

There are several approaches to imbalanced machine learning modeling. We expose the main idea of the approaches and the models analyzed in this paper, all of them were computed using the *imblearn* library (Python) except *xgboost* that has its own library (*Python*).

- **Oversampling methods.** Replicates instances from the minority class to balance the class distribution. We used a Random Over Sampler, SMOTE, and SMOTE Tomek, and ADASYN models.

- **Undersampling methods.** Randomly removes instances from the majority class to create a balanced dataset. It is considered a worse method because of the loss of information. We used a Random Under Sampler model.

- **Bagging methods.** It trains multiple models on different bootstrap samples of the imbalanced dataset and then combines their predictions averaging. These models are good because they reduce overfitting and improve the performance of imbalanced learning tasks. We used a Balanced Bagging Classifier model and a Balanced random forest.

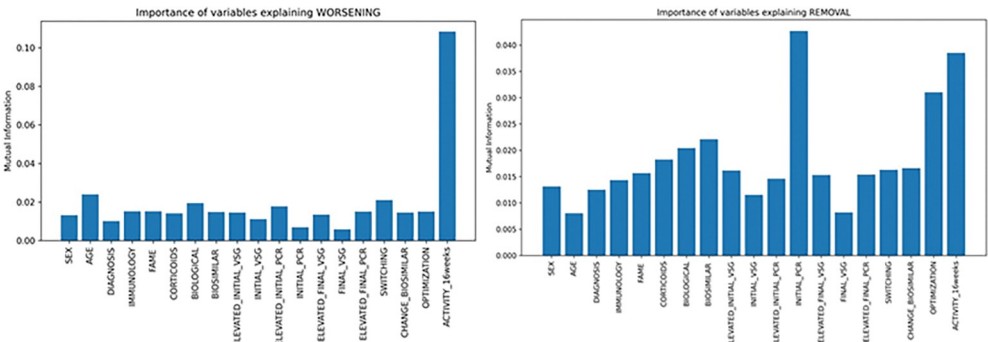

**Fig 1. Mutual information test to compute the most relevant variables explaining the increase in disease activity in six months.**

**Table 1. Comparison of the metrics: F1-score, mean ROC AUC, and accuracy of the different imbalanced ML models to predict "worsening".**

| Imbalanced ML Model | F1-score | Mean ROC AUC | Accuracy |
|---|---|---|---|
| Random under sampler | 0.45 | 0.76 | 0.78 |
| Random over sampler | 0.52 | 0.83 | 0.76 |
| SMOTE | 0.48 | 0.89 | 0.76 |
| SMOTE Tomek | 0.45 | 0.82 | 0.75 |
| Weighted logistic regression | 0.47 | 0.82 | 0.71 |
| Balanced Bagging Classifier | 0.51 | 0.82 | 0.81 |
| Weighted XGBoost | 0.45 | 0.69 | 0.8 |
| Balanced Random Forest | 0.47 | 0.80 | 0.71 |
| ADASYN | 0.46 | 0.47 | 0.81 |

- **Weighted methods.** They assign higher weights to the minority class to produce more significance during the model fitting process. We used a weighted logistic regression model and a weighted xgboost model. The last one is also good because gradient-boosting methods, that are good improving the performance of the minority class during the training process

We compare the previous imbalanced ML models in **Tables 1 and 2** we use three different metrics. By order of importance:

- **F1-score** of the class 1 of the target variable, is a good metric because since it is the harmonic mean of precision and recall it balances both to detect the increase of disease activity.

- **Mean ROC AUC**. It plots the true positive rate (recall) against the false positive rate, it considers the model's performance across various decision thresholds and is not heavily influenced by class imbalance, which is why it is less sensitive to class imbalance than accuracy or precision.

- **Accuracy.** It is the traditional metric of non-imbalanced ML models. In imbalanced ML models with similar F1-score and mean ROC AUC, we will take those with the highest accuracy in the prediction.

Once the best models have been selected, we will compute the confusion matrices and the corresponding ROC curves. If feasible, we will also visualize the decision tree to make the model explicit and facilitate an assessment of the variables' roles (see **Figs 2–4**), employing the libraries *matplotlib* and *seaborn* (*Python*).

**Table 2. Comparison of the metrics: F1-score, mean ROC AUC, and accuracy of the different imbalanced ML models to predict "removal".**

| Imbalanced ML Model | F1-score | Mean ROC AUC | Accuracy |
|---|---|---|---|
| Random under sampler | 0.56 | 0.75 | 0.69 |
| Random over sampler | 0.59 | 0.66 | 0.69 |
| SMOTE | 0.58 | 0.65 | 0.67 |
| SMOTE Tomek | 0.63 | 0.74 | 0.74 |
| Weighted logistic regression | 0.51 | 0.67 | 0.43 |
| Balanced Bagging Classifier | 0.5 | 0.54 | 0.63 |
| Weighted XGBoost | 0.56 | 0.63 | 0.64 |
| Balanced Random Forest | 0.54 | 0.66 | 0.58 |
| ADASYN | 0.5 | 0.54 | 0.6 |

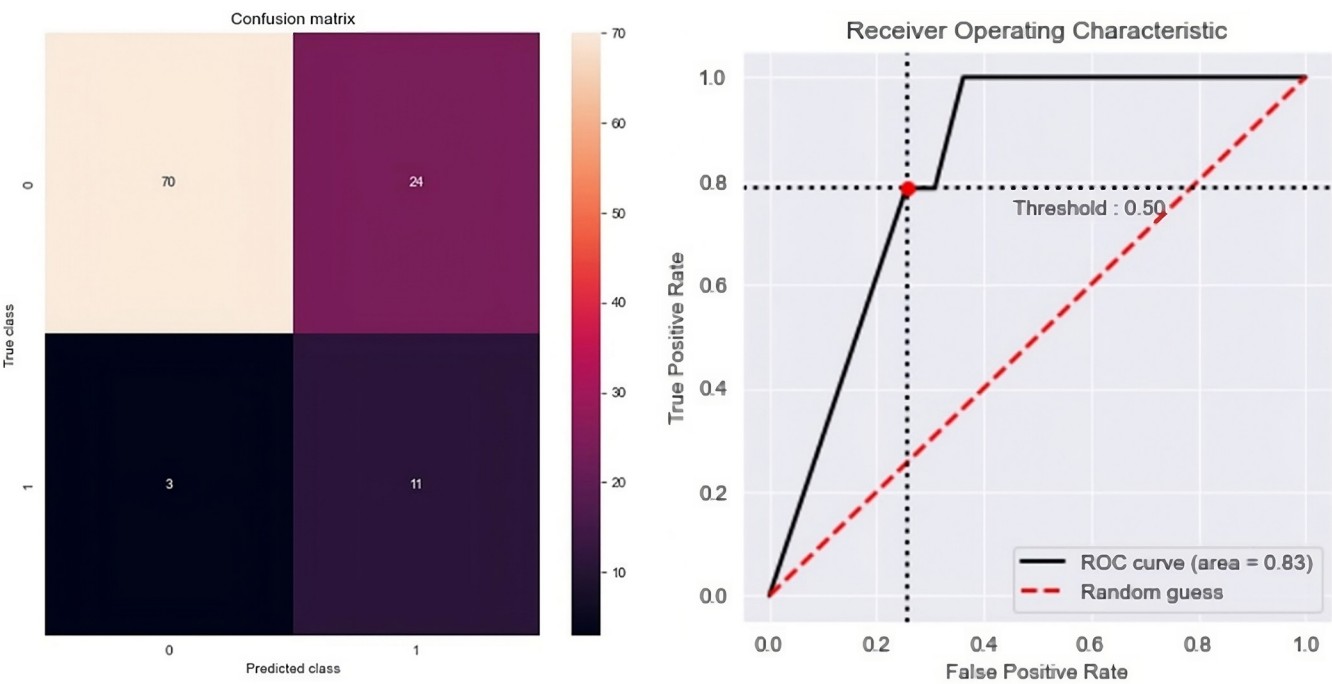

**Fig 2. Confusion matrix and ROC curve of the ROS model.**

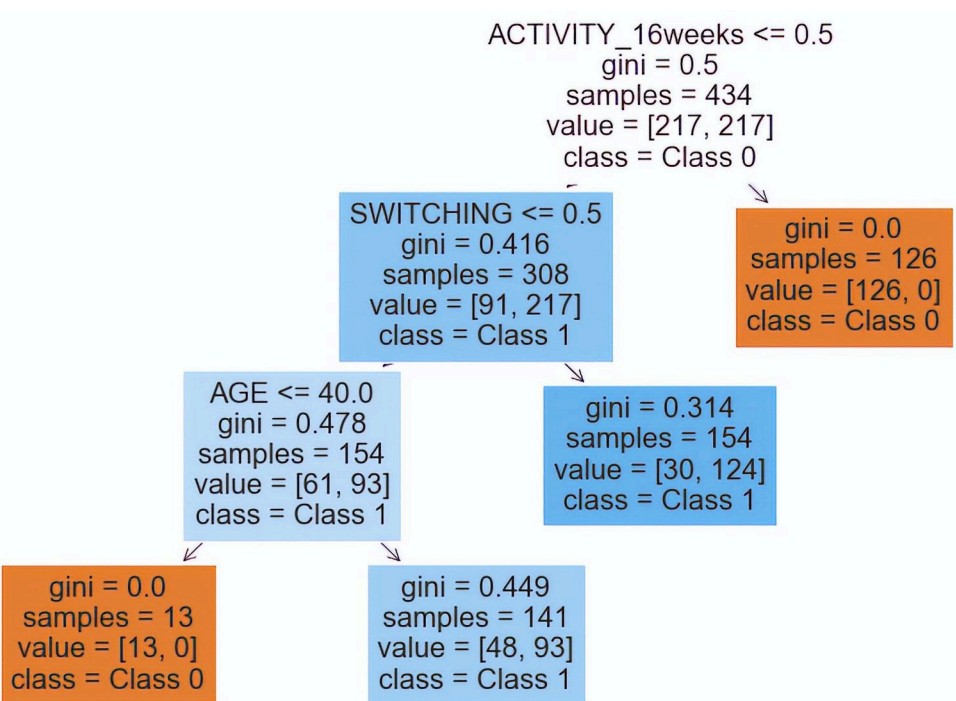

**Fig 3. Decision tree of worsening.** Model ROS.

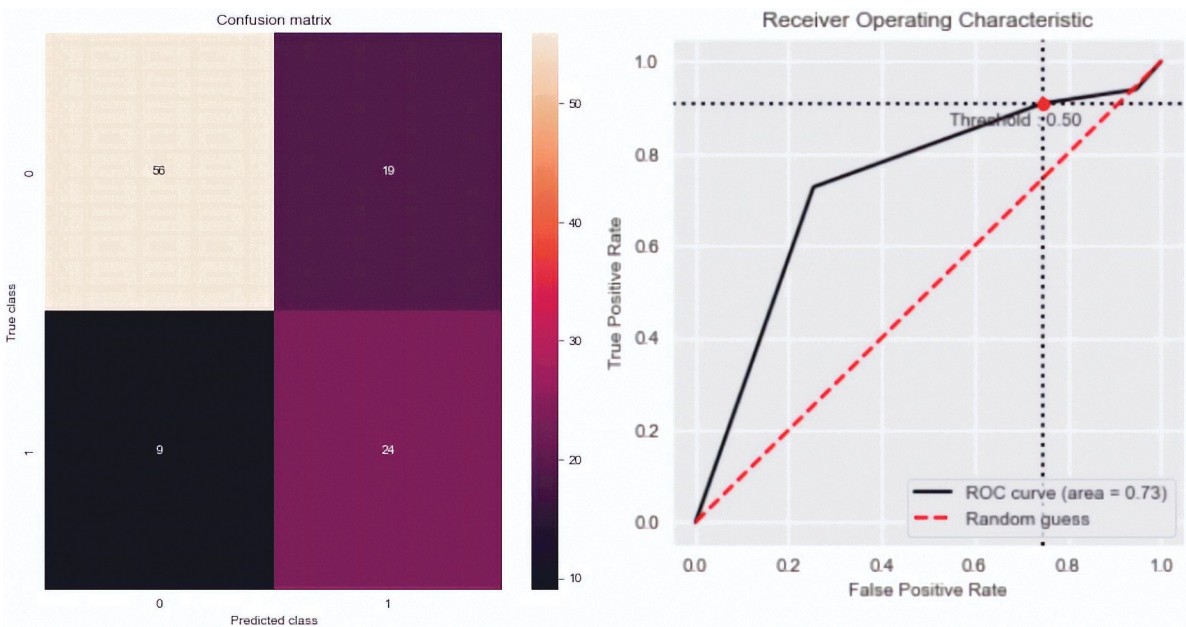

**Fig 4. Confusion matrix and ROC curve of the SMOTE Tomek model.**

### Ethical approval information

We have obtained final approval from the Clinical Research Ethics Committee of the General University Hospital of Ciudad Real, which was granted on October 25, 2022 (act 10/2022, C-567). Additionally, we obtained written informed consent from the patients to publish the material.

Moreover, the authors did not have access to information that could identify individual participants during or after data collection.

## Results

Of the 380 patients being treated with biosimilar bDMARDs, a total of 364 patients who met the inclusion criteria were selected (3 did not meet the inclusion criteria, and 13 were lost to follow-up). The mean age was 52.50 years (± 12.11), with 168 women and 196 men included in the study. In terms of the number of patients, the drugs used were: 203 adalimumab, 130 etanercept, 13 infliximab, and 18 rituximab. Among the total patients, 173 had spondyloarthritis, 68 had psoriatic arthritis, 112 had rheumatoid arthritis (90 seropositive and 22 seronegative), and 11 had other systemic autoimmune diseases (Behçet's disease, systemic lupus erythematosus, Sjögren's syndrome, dermatomyositis, and Papash syndrome).

In the mutual information test (**Fig 1**), we obtained that the most important features explaining the worsening of the disease were: "activity (16 weeks)", "age", "switching", "biological", and "elevated initial PCR". The variable "activity" exhibited a noteworthy level of significant mutual information independently. Since it is correlated inversely proportional with worsening, it may remain undetected if we had used traditional feature selection tests, such as the F-test or chi-square test. The rest of the mentioned variables affect the target working together. As far as removing biosimilars is concerned, the most important features were: "elevated initial PCR", "Activity (16 weeks)", "optimization", "biological", and "biosimilar", but these last two were dropped performing a PCA analysis. In this case, there is not a significant variable explaining the target alone, but mixing them.

After computing the principal metrics of the different imbalanced ML models to explain "worsening" (**Table 1**), we selected the random over sampler model, because it has the highest f1-score (0.52), the second highest Mean ROC AUC (0.83) and a good accuracy (0.76). The confusion matrix and the ROC curve of this model can be consulted in **Fig 2,** where we can appreciate that our model detects 11/14 patients with worsening disease and that the ROC AUC is 0.83. This model generated a decision tree, **Fig 3**, with the following interpretation: the patients with worsening disease after six months are:

- Patients with no activity at 16 weeks, that have switched the biosimilar

- Patients with no activity at 16 weeks, without switching, and over 40 years old.

Therefore, the percentage of patients who worsen after 24 weeks is 13.13%.

As far as the target variable "removal" is concerned, we chose the Smote Tomek imbalanced model because it has the highest f1-score (0.63), the second highest mean ROC AUC (0.74), and good accuracy (0.74). The confusion matrix and the ROC curve of this model can be consulted in **Fig 4,** where this model detects 24/33 removals of the biosimilar and the ROC AUC is 0.73 with a pronounced imbalance toward 1 (see the optimal operating point, the red point in Fig 4). As in the previous case, this model is also interpretable and generates a decision tree, **Fig 5**, with the following meaning: the removal of the biosimilar is more probable in:

- Patients with no optimization, with initial PCR <1.424, and with disease activity at 16 weeks.

Therefore, the percentage of patients who remove the biosimilar drug at 24 weeks is 30.45%.

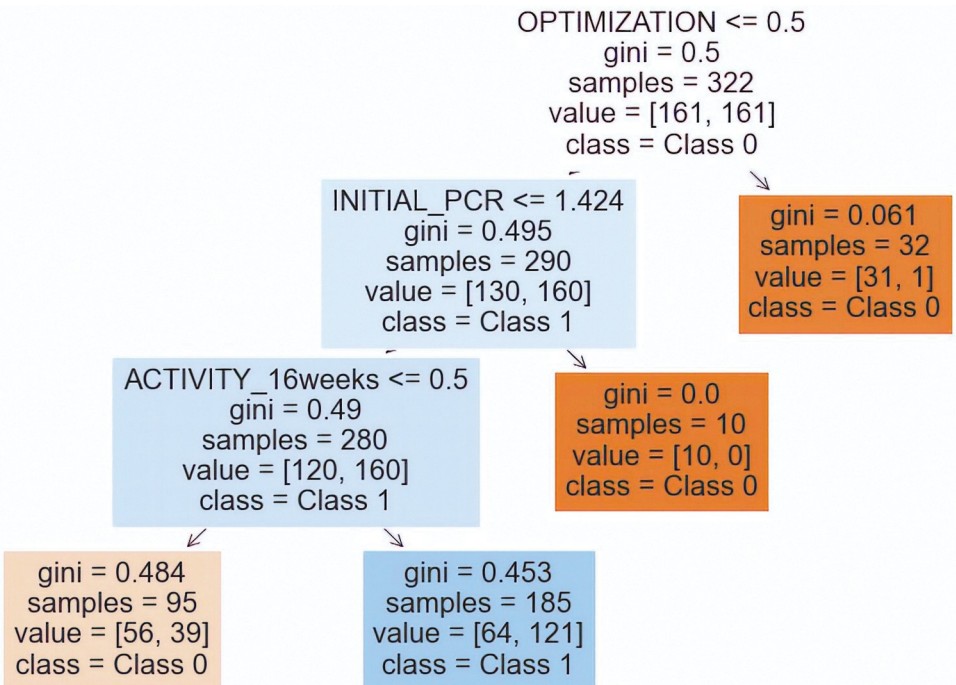

**Fig 5. Decision tree of removal.** Model SMOTE Tomek model.

## Discussion

Our study is a real-life practice study of biosimilar bDMARDs, conducted in a large patient population. It was observed that 29.95% of the participants had to discontinue the biosimilar drug, mainly due to its lack of efficacy, which exceeds the average reported in the current literature, such as the Glintborg study, which reported only 7% [11]. Only 18 patients experienced some adverse effects, of which only 2 cases were severe, a slightly lower number than in the Bruni study (4.74% in our research vs. 22.73%) [12]. Biosimilar drugs were effective and did not show significant interference in inflammatory activity. However, there are cases where the medication must be withdrawn, or the patient experiences a severe worsening of their condition due to the use of biosimilars. These cases, though infrequent, should be detected using all the tools that artificial intelligence offers, just as it is used for the detection of cancer and other abnormal medical cases.

When comparing machine learning imbalanced methodology with traditional statistical approaches, several advantages emerge concerning complexity, accuracy, and specificity in detecting the imbalanced class. For instance, let us consider the variable "activity at 16 weeks" and a logit regression classical model. Initially, there is no relationship between "activity at 16 weeks" and "removal" or "worsening" in a classical context (chi-square or F-test). Furthermore, in the classical logit model, the p-value for this variable is almost 1, indicating no statistical significance ($p > 0.05$). Despite employing weighted logistic regression with different class weights during the training process (as shown in Tables 1 and 2), the imbalanced model performs worse than the selected models, and the variable "activity at 16 weeks" still has a p-value $> 0.05$.

Therefore, patients with worsening disease after 6 months are patients with no activity at 16 weeks that have switched the biosimilar, and, patients with no activity at 16 weeks without switching and over 40 years old.

Moreover, the removal of the biosimilar is more probable in patients with no optimization with initial PCR $<1.424$ mg/dl and with disease activity at 16 weeks.

As suggestions for improvement, we propose strengthening the foundation of our study through the following approaches:

- Long-term patient follow-up: Conducting a long-term follow-up of patients can provide valuable insights into the progression of the disease and the effectiveness of treatments over extended periods. This longitudinal analysis can help establish stronger correlations and identify patterns that may not be evident in shorter-term studies.

- Increased sample size: Expanding the sample size of the study can enhance the Machine Learning imbalanced power and generalizability of the findings. A larger sample allows for more robust conclusions and a better representation of the target population, minimizing the risk of biased results.

- Cross-hospital collaboration: Collaborating with other hospitals or medical institutions can enrich the study's dataset and improve the diversity of patient cases. Sharing data and information across institutions can lead to a more comprehensive analysis, capturing a broader range of patient demographics and medical conditions.

## Conclusion

In this paper, we present an imbalanced machine learning methodology, common in data science but not previously employed to analyze the behavior of biosimilar medications in rheumatic diseases. Using mutual information as feature selection and choosing the best-

imbalanced model as the one with the best f1-score for the imbalanced class and with good mean ROC AUC and accuracy, we discovered two decision trees with considerably high precision metrics, which explain our target variables: patient worsening (Fig 3) and biosimilar removal (Fig 5). Additionally, we identified variables that were not previously considered in the literature, such as age in the patient worsening model and switching and optimization in the biosimilar removal model.

## Supporting information

**S1 File.**
(PDF)

**S2 File.**
(PDF)

## Author Contributions

**Conceptualization:** David Castro Corredor, Luis Ángel Calvo Pascual.

**Investigation:** David Castro Corredor, Luis Ángel Calvo Pascual.

**Methodology:** David Castro Corredor, Luis Ángel Calvo Pascual.

**Project administration:** David Castro Corredor.

**Supervision:** David Castro Corredor.

**Validation:** David Castro Corredor, Luis Ángel Calvo Pascual.

**Writing – original draft:** David Castro Corredor, Luis Ángel Calvo Pascual.

**Writing – review & editing:** David Castro Corredor.

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
