## [Decision Letter · Decision Letter 0]

26 Sep 2023

PONE-D-23-27865IMBALANCED MACHINE LEARNING CLASSIFICATION MODELS FOR REMOVAL BIOSIMILAR DRUGS AND INCREASED ACTIVITY IN PATIENTS WITH RHEUMATIC DISEASESPLOS ONE

Dear Dr. Castro Corredor,

Thank you for submitting your manuscript to PLOS ONE. After careful consideration, we feel that it has merit but does not fully meet PLOS ONE’s publication criteria as it currently stands. Therefore, we invite you to submit a revised version of the manuscript that addresses the points raised during the review process.

We look forward to receiving your revised manuscript.

Kind regards,

Wenguo Cui, Ph.D

Academic Editor

PLOS ONE

Journal Requirements:

5. Please include your tables as part of your main manuscript and remove the individual files. Please note that supplementary tables (should remain/ be uploaded) as separate "supporting information" files

Reviewers' comments:

Reviewer's Responses to Questions

**Comments to the Author**

1. Is the manuscript technically sound, and do the data support the conclusions?

Reviewer #1: Yes

Reviewer #2: Yes

2. Has the statistical analysis been performed appropriately and rigorously? 

Reviewer #1: Yes

Reviewer #2: I Don't Know

3. Have the authors made all data underlying the findings in their manuscript fully available?

Reviewer #1: Yes

Reviewer #2: Yes

4. Is the manuscript presented in an intelligible fashion and written in standard English?

Reviewer #1: Yes

Reviewer #2: Yes

5. Review Comments to the Author

Reviewer #1: This is a retrospective observational study that examined patients with immune-mediated inflammatory rheumatic diseases who switched from a biological drug to a biosimilar drug. The aim was to predict long-term disease worsening and the discontinuation of biosimilar medication. The study employed imbalanced machine learning models due to the low prevalence of disease worsening cases. Two decision tree models were developed, with significant variables including switching of biosimilar, age, optimization, and initial PCR value. The models achieved f1-scores of 0.52 and 0.63 for disease worsening and biosimilar removal prediction, respectively. The results suggested that biosimilar drugs may not always be effective in rheumatic diseases. The study highlights the potential of imbalanced machine learning models in identifying cases requiring drug discontinuation and detecting increased disease activity. But this article needed a great deal of revision before it could be considered ready for formal publication.

1. The introduction lacks a clear and concise research objective or aim.The introduction should clearly state the research objective or aim, such as investigating the long-term effects and outcomes of using biosimilar drugs in patients with inflammatory rheumatic diseases.

2. The limitations of traditional statistical models in dealing with imbalanced data are mentioned but not clearly explained. It would be helpful to provide a brief overview of the challenges faced by these models when handling imbalanced datasets.

3. The description of the study design is not clear and concise. It would be helpful to explicitly state the objective of the study and provide a brief overview of the methods used. Additionally, it would be beneficial to mention any specific hypotheses being tested.

4. The description of the database and patient selection process is incomplete. It is important to provide more details on the source of the database, how patients were identified and recruited, and any inclusion/exclusion criteria applied.

5. The discussion of imbalanced target variables is brief and lacks in-depth analysis. It would be beneficial to provide more information on the significance and implications of these variables, as well as potential challenges in modeling imbalanced datasets. Expand the discussion on imbalanced target variables, including their importance, impact on the study findings, and potential limitations in analyzing imbalanced datasets.

6. The discussion section needs to elaborate on the implications and limitations of the findings. It is important to provide a balanced interpretation of the results and compare them with previous studies in the field. Additionally, suggestions for future research directions or clinical implications can be included to enhance the relevance and practicality of the study.

Reviewer #2: In this paper, the authors computed the best-balanced machine learning model for predicting disease progression and biosimilar removal. But I think this article needs major revisions to re-evaluate its suitability for publication on PLOS ONE. The specific comments are listed below:

1. References are the key to readers' understanding. Please confirm the correctness of the references so that readers will be able to locate the correct documents. After that please check the manuscript thoroughly and eliminate all the lumps in the manuscript. This should be done by characterizing each reference individually This can be done by mentioning 1 or 2 phrases per reference to show how it is different from the others and why it deserves mentioning.

2. A large number of abbreviations appear in the manuscript, and these abbreviations are not well explained, causing a lot of confusion to readers. In addition, please strictly implement the corresponding full name when the abbreviation of the noun appears for the first time.

3. Please check and modify the grammar and English expression. Thanks.

4. Why use different learning models to analyze problems?

5. Please provide a higher resolution image, a lot of key data is not clearly visible.

6. PLOS authors have the option to publish the peer review history of their article (what does this mean?). If published, this will include your full peer review and any attached files.

Reviewer #1: No

Reviewer #2: No

---

## [Author Response · Author response to Decision Letter 0]

13 Nov 2023

Dear Reviewer,

We would like to express our sincere thanks for your comprehensive review of our manuscript. Following your suggestions, we have made substantial revisions to the paper. These changes have significantly enhanced the clarity and depth of our work.

Journal Requirements:

1. We have made a series of changes so that the manuscript meets the PLOS One-style requirements.

2. The online version contains supplementary material available at https://doi.org/10.5281/zenodo.10072867 (in the section “supporting information”)

3. The funders had no role in study design, data collection and analysis, publication decisions, or manuscript preparation. In addition, we clarify that the authors received no specific funding for this work. 

4. The minimal data set underlying the results described in your manuscript can be found at Zenodo at https://doi.org/10.5281/zenodo.10072867.

5. The tables are included as part of the main manuscript. 

Reviewers' comments. Reviewer's Responses to Questions. Comments to the Author:

Reviewer #1: 

1. The introduction lacks a clear and concise research objective or aim.The introduction should clearly state the research objective or aim, such as investigating the long-term effects and outcomes of using biosimilar drugs in patients with inflammatory rheumatic diseases.

Now the fundamental objective has a separate paragraph, the fourth in the introduction, lines 64-71. It emphasizes the importance of understanding "worsening" for risk assessment and "removal" for evaluating drug tolerance and effectiveness, impacting clinical, regulatory, and financial decisions in rheumatology

2. The limitations of traditional statistical models in dealing with imbalanced data are mentioned but not clearly explained. It would be helpful to provide a brief overview of the challenges faced by these models when handling imbalanced datasets.

This is explained in the second paraghraph of the introduction lines 37-50 and in the discussion lines 239-245. Basically, traditional methods make restrictive assumptions that don't always apply to complex disease data, leading to potential inaccuracies, especially with imbalanced datasets where they tend to favor the majority class. This results in biased probability estimates and misses nuanced patterns critical in rare conditions.

3. The description of the study design is not clear and concise. It would be helpful to explicitly state the objective of the study and provide a brief overview of the methods used. Additionally, it would be beneficial to mention any specific hypotheses being tested.

This is detailed in sections Study Design and Data analysis (lines 137-194). The study monitors the progression of rheumatic diseases by evaluating disease activity at the 16-week mark (initial activity) and again at 24 weeks (final activity), using markers such as CRP and ESR levels. It focuses on two key outcomes: "worsening," where disease activity notably increases, observed in 13.13% of patients, and "removal," where a biosimilar is discontinued due to inefficacy or adverse effects, noted in 30.45% of patients. The research aims to forecast these specific outcomes to tailor treatment plans more effectively. Data analysis involves using the mutual information test to discern important variables and applying robust metrics like the f1-score across various machine learning models suited for imbalanced data. We mention as specific hypotheses in the introduction the effects of comorbidities like age, obesity, etc.

4. The description of the database and patient selection process is incomplete. It is important to provide more details on the source of the database, how patients were identified and recruited, and any inclusion/exclusion criteria applied.

It is described in the section Patients. The inclusion and exclusion criteria are described in the second paragraph (lines 91-99). We admitted all patients over the age of 18 who arrived at the department and signed the consent form. In our opinion, there is no bias since no patient was excluded.

5. The discussion of imbalanced target variables is brief and lacks in-depth analysis. It would be beneficial to provide more information on the significance and implications of these variables, as well as potential challenges in modeling imbalanced datasets. Expand the discussion on imbalanced target variables, including their importance, impact on the study findings, and potential limitations in analyzing imbalanced datasets.

We have provided a clearer explanation of the target variables in the variables section and have added lines 247-252 in the discussion. Most of the literature emphasizes that biosimilar drugs are just as effective as the originals, so few studies investigate the exacerbation of activity or why treatment is discontinued. We mention the hypothesis of the nocebo effect, but we consider our two trees as another explanation.

6. The discussion section needs to elaborate on the implications and limitations of the findings. It is important to provide a balanced interpretation of the results and compare them with previous studies in the field. Additionally, suggestions for future research directions or clinical implications can be included to enhance the relevance and practicality of the study.

In the discussion section, we clarify that our decision trees offer a probabilistic causal analysis for the target variables, yet this explanation is not without doubt, as detailed in lines 253-268. We continue to emphasize the enhancements in lines 259-268, noting that the analysis of imbalanced machine learning tends to improve with a larger patient sample size.

Reviewer #2: 

1. References are the key to readers' understanding. Please confirm the correctness of the references so that readers will be able to locate the correct documents. After that please check the manuscript thoroughly and eliminate all the lumps in the manuscript. This should be done by characterizing each reference individually This can be done by mentioning 1 or 2 phrases per reference to show how it is different from the others and why it deserves mentioning.

We have improved the references, which were previously too generic and are now more specific. Each cited paper discusses the preceding sentence to support it.

2. A large number of abbreviations appear in the manuscript, and these abbreviations are not well explained, causing a lot of confusion to readers. In addition, please strictly implement the corresponding full name when the abbreviation of the noun appears for the first time.

We have explained all the acronyms and abbreviations throughout the text, to mention a few examples: Biological Disease-Modifying Antirheumatic Drugs (bDMARDs), ROC (Receiver Operating Characteristic), Area Under the Curve (AUC), ASAS (Assessment of SpondyloArthritis International Society), EULAR (European League Against Rheumatism), DMARDs (Disease-Modifying Antirheumatic Drugs), Erythrocyte Sedimentation Rate (ESR), C-reactive protein (CRP), Ankylosing Spondylitis Activity Score (ASDAS), Synthetic Minority Over-sampling Technique (SMOTE), Adaptive Synthetic Sampling (ADASYN).

3. Please check and modify the grammar and English expression. Thanks.

We have tried to improve the grammar throughout the text by making changes to provide greater clarity and naturalness to the text.

4. Why use different learning models to analyze problems?

Explained in Section Data Analysis lines: 175-182. It improves analysis by combining model abilities (ones works better with linear/non linear), balancing simple and complex pattern recognition, reducing overfitting, and minimizing bias, leading to more accurate results.

5. Please provide a higher resolution image, a lot of key data is not clearly visible.

We have doubled the resolution of the 5 images.

Thanks again and best regards,

The authors

---

## [Editor Report · Decision Letter 1]

15 Nov 2023

Imbalanced machine learning classification models for removal of biosimilar drugs and increased activity in patients with rheumatic diseases

PONE-D-23-27865R1

Dear Dr. David Castro Corredor,

We’re pleased to inform you that your manuscript has been judged scientifically suitable for publication and will be formally accepted for publication once it meets all outstanding technical requirements.

Kind regards,

Wenguo Cui, Ph.D

Academic Editor

PLOS ONE
---

## [Editor Report · Acceptance letter]

22 Nov 2023

PONE-D-23-27865R1 

Imbalanced machine learning classification models for removal biosimilar drugs and increased activity in patients with rheumatic diseases 

Dear Dr. Castro Corredor:

I'm pleased to inform you that your manuscript has been deemed suitable for publication in PLOS ONE. Congratulations! Your manuscript is now with our production department. 

Kind regards, 

on behalf of

Professor Wenguo Cui 

Academic Editor

PLOS ONE